# DATASET CURATION BEYOND ACCURACY

## ABSTRACT

Neural networks are known to be data-hungry, and collecting large labeled datasets is often a crucial step in deep learning deployment. Researchers have studied dataset aspects such as distributional shift and labeling cost, primarily using downstream prediction accuracy for evaluation. In sensitive real-world applications such as medicine and self-driving cars, not only is the accuracy important, but also the calibration – the extent that model uncertainty reflects the actual correctness likelihood. It has recently been shown that modern neural networks are ill-calibrated. In this work, we take a complementary approach – studying how dataset properties, rather than architecture, affects calibration. For the common issue of dataset imbalance, we show that calibration varies significantly among classes, even when common strategies to mitigate class imbalance are employed. We also study the effects of label quality, showing how label noise dramatically increases calibration error. Furthermore, poor calibration can come from small dataset sizes, which we motive via results on network expressivity. Our experiments demonstrate that dataset properties can significantly affect calibration and suggest that calibration should be measured during dataset curation.

## 1 INTRODUCTION

Neural networks often require large amounts of labeled data to perform well, making data curation a crucial but costly aspect of deployment. Thus, researchers have studied dataset properties such as distributional shift (Miller et al., 2020) and the bias in crowd-sourced computer vision datasets (Tsipras et al., 2020) among others. Often, the evaluation criteria in such studies is downstream prediction accuracy. However, neural networks are increasingly deployed in sensitive real-world applications such as medicine (Caruana et al., 2015), self-driving cars (Bojarski et al., 2016), and scientific analysis (Attia et al., 2020), where not only accuracy matters but also calibration. Calibration is the extent to which model certainty reflects the actual correctness likelihood. Calibration can be important when costs of false positives and false negatives are asymmetric, e.g., for a deadly disease with cheap treatment, doctors might initiate treatment when the probability of being sick exceeds $10\%$. Beyond simple classification, calibration can be important for beam-search in NLP (Ott et al., 2018) and algorithmic fairness (Pleiss et al., 2017). Calibration in machine learning has been studied by e.g Zadrozny & Elkan (2001); Naeini et al. (2015). Niculescu-Mizil & Caruana (2005) have shown that small scale neural networks can yield well-calibrated predictions. However, it has recently been observed by (Guo et al., 2017) that modern neural networks are ill-calibrated, whereas the now primitive Lenet (LeCun et al., 1998) achieves good calibration.

In this work, we take a complementary approach; instead of focusing on network architecture, we study how calibration is influenced by dataset properties. We primarily focus on computer vision and perform extensive experiments across common benchmarks and more exotic datasets such as satellite images (the eurosat dataset (Helber et al., 2019)) and species detection (the iNaturalist dataset (Van Horn et al., 2018)). We consistently find that dataset properties can significantly affect calibration, causing effects comparable to network architecture. For example, we consider the ubiquitous problem of class imbalanced datasets, a common issue in practice (Van Horn et al., 2018; Krishna et al., 2017; Thomee et al., 2016). For such datasets, the miscalibration is not uniform but instead varies across the different classes. This problem persists even when common strategies to mitigate class imbalanced are employed. Another practical concern is generating high-quality labels via e.g. crowdsourcing (Karger et al., 2011). We demonstrate how labeling quality affects calibration, with noisier labels resulting in worse calibration. Additionally, we show that just the size of the dataset

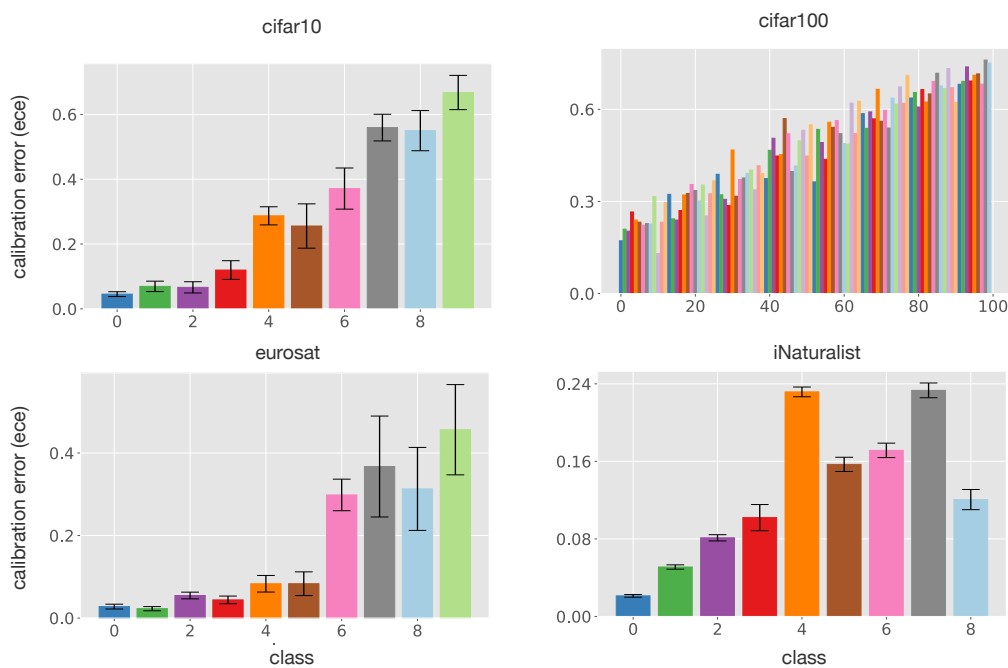

Figure 1: Calibration error for individual classes under class-imbalance. The classes are ordered from the most (left) to the least (right) amount of samples. Fewer samples result in larger calibration errors. Imbalance is injected in CIFAR10/100 and eurosat randomly, removing any correlation with class-specific properties. We do not modify Inaturalist, which already suffers from imbalance; thus classwise calibration is correlated with class-specific properties.

has a strong effect on calibration. This also holds when one artificially increases the dataset size by data augmentation. We motivate our findings by considering the geometry of the cross-entropy loss and utilizing recent results on network expressivity (Yun et al., 2019). If the dataset is sufficiently small compared to the number of parameters, we argue that the lack of minimizer for the cross-entropy loss biases the network to high confidence and poor calibration. Our results highlight an underappreciated aspect of calibration and suggest that for sensitive applications, one should measure calibration during dataset curation.

## 2 BACKGROUND

**Calibration.** Calibration has a traditional place in machine learning (Zadrozny & Elkan, 2001; Naeini et al., 2015). Before the advent of modern deep learning, Niculescu-Mizil & Caruana (2005) showed that neural networks can yield well-calibrated predictions for classification. However, Guo et al. (2017) showed that *modern* neural networks are ill-calibrated. Modern neural networks are modeled as e.g. resnet (He et al., 2016) or densenets (Huang et al., 2016). It is important to note that accuracy and calibration do *not* necessarily follow each other, but can move independently – modern neural networks are ill-calibrated, but still yield excellent accuracy. Beyond image classification, the importance of calibration in NLP has further been studied by Ott et al. (2018) and its relationship to fairness by Pleiss et al. (2017).

**Metrics for Calibration.** We let $\{x_i\} \in \mathbb{R}^{n \times d_x}$ be a dataset of $n$ datapoints with $d_x$ features and take $\{y_i\}$ to be the labels. Following Guo et al. (2017), we assume that a neural network $h$ outputs $h(x_i) = (\hat{p}_i, \hat{y}_i)$, where $\hat{y}_i$ is the predicted class and $\hat{p}_i$ is the estimated probability that the prediction is correct. For evaluating calibration, we divide the interval $[0, 1]$ into $M$ equally sized bins and assign predictions to bins based upon $\hat{p}$. Within each bin $B_m$ we define the accuracy as $\text{acc}(B_m) = \frac{1}{|B_m|} \sum_{i \in B_m} \mathbf{1}(\hat{y}_i = y_i)$. Similarly, we define the confidence as $\text{conf}(B_m) = \frac{1}{|B_m|} \sum_{i \in B_m} \hat{p}_i$. For a well-calibrated model, we would expect the confidence and accuracy of each bin to be close to each other. Calibration error can be measured by their difference, evaluated on the test set. The

resulting metric is known as the expected calibration error (Naeini et al., 2015), often abbreviated as ece. Mathematically, it is defined as follows:

$$\text{ece} = \sum_{m=1}^{M} \frac{|B_m|}{n} \left| \text{acc}(B_m) - \text{conf}(B_m) \right| \qquad (1)$$

## 3 EXPERIMENTS

**Experimental setup.** We consider the following computer vision datasets. **Cifar10 & Cifar100** (Krizhevsky & Hinton, 2010) which contains 50,000 RGB images spanning ten or hundred classes respectively. Classes are balanced. **Eurosat** (Helber et al., 2019), which is a dataset of satellite images over continental Europe; there are ten balanced classes and 27,000 images in total. **iNaturalist** (Van Horn et al., 2018) which is a dataset for species detection. We use the FGVC6 version (FGVC6, 2019), compromising over 260,000 images and an imbalanced hierarchical class system compromising e.g., species and phylum. We perform classification at the "class" level, resulting in nine classes. Across all datasets, we use the same architecture, Resnet50 (He et al., 2016). We use hyperparameters from the original resnet paper (He et al., 2016): cross-entropy loss optimized with SGD using a learning rate at $0.1$ and decreased by a factor $0.1$ after $50\,\%$ and $75\,\%$ of the training, a batch size of $128$, a weight decay of $0.0001$, and momentum of $0.9$. For the cifar/eurosat/inaturalist, networks are trained over $62/30/331 \times 10^3$ gradient steps, corresponding to 160 epochs for each dataset. We use randomized cropping and random horizontal flipping for data augmentation, see Appendix A for data preprocessing. Experiments are repeated five times, with mean and standard deviations reported. Calibration error is measured in expected calibration error(ece) as in eq. (1), using $M = 15$ and evaluated on the test set.

**Inbalanced dataset.** A common problem in practice, not necessarily found in benchmark datasets, is class imbalance (i.e., the number of available samples varies between classes), see Van Horn et al. (2018); Krishna et al. (2017); Thomee et al. (2016). Here we study how imbalanced datasets affect

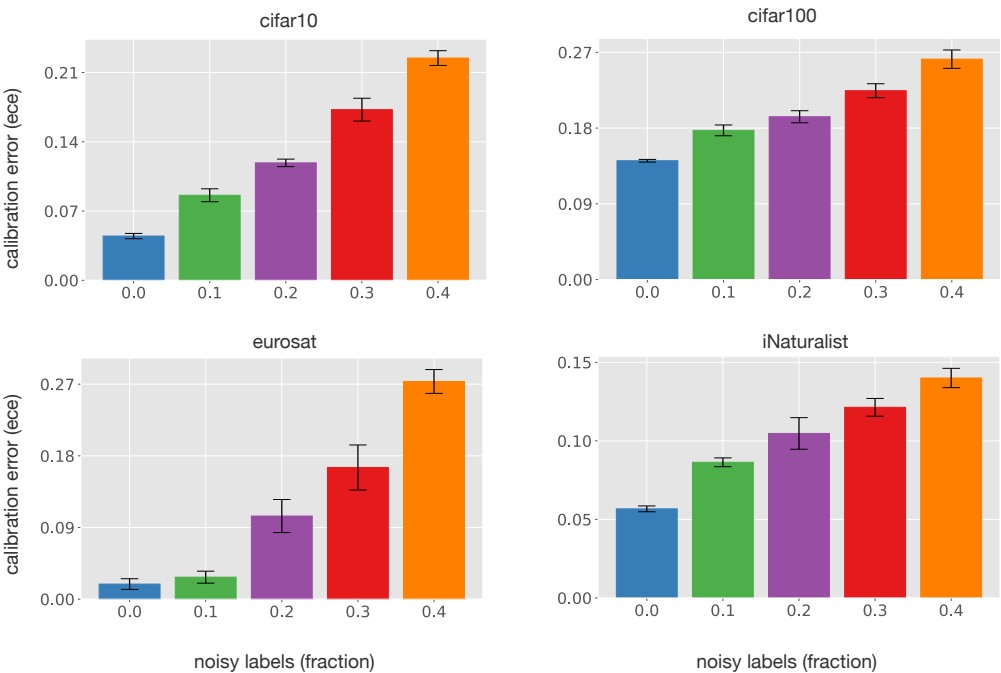

Figure 2: Calibration error under label noise, simulated by randomly reassigning labels for a fraction of the training labels. Across datasets, label noise degrades network calibration. Thus, label noise from e.g. crowd sourcing can affect not only accuracy, but also calibration (Karger et al., 2011).

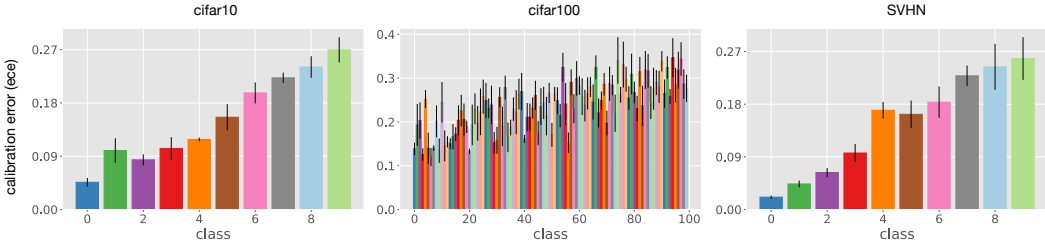

Figure 3: Calibration error under non-uniform label noise. We linearly increase label noise from 0 to 0.5 among classes, and sort them thereafter. Increased noise leads to worse calibration.

the calibration error for individual classes. Whereas the iNaturalist dataset is naturally imbalanced, the cifar and eurosat datasets are not. For these datasets, we simulate long-tailed class imbalance following Cao et al. (2019). We randomly reorder the classes from $= 1$ to $n$, and only keep a $\alpha^{i-1}$ fraction of examples for class $i$. Given some desired ratio $\rho$ between the class with the most and fewest samples, one picks $\alpha$ such that $\alpha^{n-1} = \rho$. Following Buda et al. (2018), we also consider a step-imbalance, where half of the classes are downsampled by a factor $\rho$. We consider $\rho = 100$ as done by e.g. Cao et al. (2019) and keep the test set balanced. For cifar/eurosat, we randomly chose what classes to subsample to eliminate class-specific properties. Since iNaturalist is already imbalanced, we keep it as it is, but note that the class-specific properties are correlated with class-specific imbalance. After this procedure, we train the DNNs as normal and give the average calibration error for individual classes. The results are shown in Figure 1. Generally, classes with fewer examples have significantly higher ECE, showing how imbalance can have a significant effect on model calibration. For the iNaturalist dataset, we have some outlier classes which is likely due to class-specific effects, e.g., the class with the most labels might be unusually hard to calibrate.

**Methods for Imbalanced Datasets.** As class imbalanced is a problem of practical importance, there is ample work on mitigating this issue. One common strategy is to sample the dataset unevenly when generating mini-batches, attempting to obtain a roughly balanced dataset. One can both oversample minority classes (Buda et al., 2018) and undersample the majority class (Japkowicz & Stephen, 2002). Another strategy is instead to weight the objective function to give all classes

Table 1: Calibration error for various mitigation strategies used in imbalanced datasets. We give the calibration error for the class with the most/fewest labels (referred to as min/max), and the ratio of these two errors. Two types of imbalance are considered, exponential and step. While improving in some cases, classwise imbalance remains even when mitigation strategies are used.

| exp-inbalance | cifar10 | | | cifar100 | | | eurosat | | |
|---|---|---|---|---|---|---|---|---|---|
| method | min | max | ratio | min | max | ratio | min | max | ratio |
| original | 0.12 | 0.48 | 4.24 | 0.34 | 0.61 | 1.82 | 0.05 | 0.3 | 7.06 |
| sampling | 0.16 | 0.62 | 3.99 | 0.39 | 0.66 | 1.68 | 0.05 | 0.22 | 4.85 |
| weighted | 0.1 | 0.35 | 3.67 | 0.22 | 0.4 | 1.79 | 0.08 | 0.14 | 1.75 |
| label smooth | 0.08 | 0.35 | 4.31 | 0.16 | 0.27 | 1.73 | 0.1 | 0.29 | 2.96 |
| focal (Lin et al., 2017) | 0.1 | 0.46 | 4.88 | 0.29 | 0.56 | 1.91 | 0.06 | 0.28 | 4.5 |
| CB (Cui et al., 2019) | 0.09 | 0.34 | 3.98 | 0.2 | 0.32 | 1.57 | 0.08 | 0.14 | 1.84 |
| step-imbalance | cifar10 | | | cifar100 | | | eurosat | | |
| method | min | max | ratio | min | max | ratio | min | max | ratio |
| original | 0.04 | 0.63 | 16.57 | 0.16 | 0.73 | 4.6 | 0.02 | 0.43 | 19.62 |
| sampling | 0.06 | 0.77 | 13.17 | 0.19 | 0.74 | 3.85 | 0.02 | 0.43 | 19.94 |
| weighted | 0.08 | 0.34 | 4.62 | 0.15 | 0.3 | 2.01 | 0.1 | 0.18 | 1.94 |
| label smooth | 0.08 | 0.47 | 5.58 | 0.12 | 0.48 | 3.95 | 0.09 | 0.33 | 3.47 |
| focal (Lin et al., 2017) | 0.03 | 0.56 | 20.54 | 0.14 | 0.66 | 4.77 | 0.03 | 0.37 | 16.72 |
| CB (Cui et al., 2019) | 0.07 | 0.41 | 6.24 | 0.15 | 0.3 | 2.04 | 0.08 | 0.17 | 2.22 |

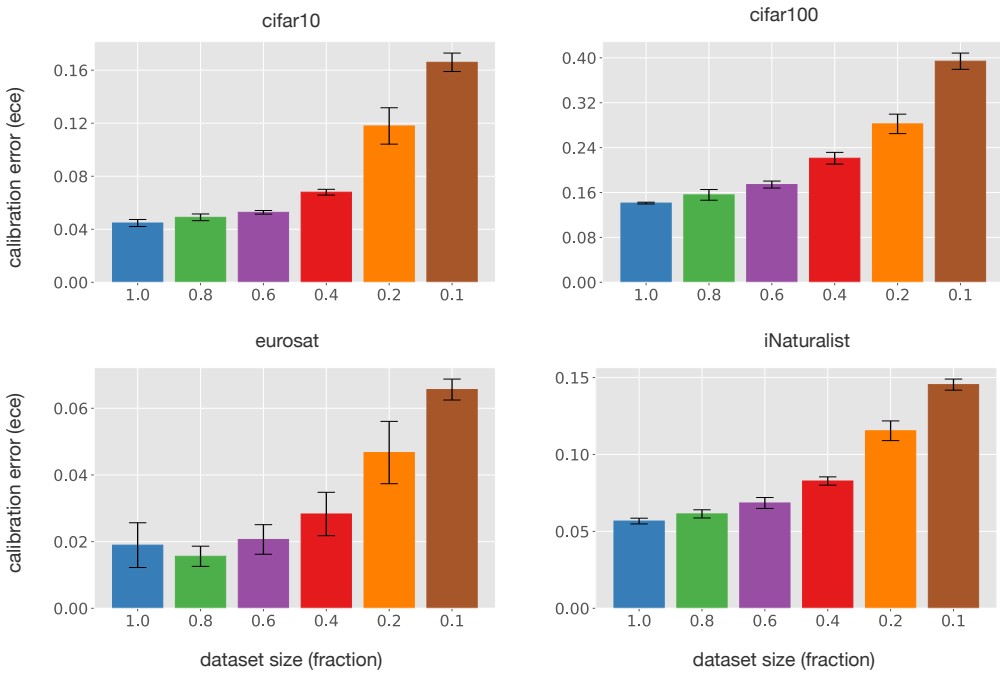

Figure 4: Calibration error under different dataset sizes. We subsample the datasets, and give the size as a fractions of the original size. Across all tasks, smaller datasets consistently yield poorer calibration, highlighting how dataset size influences not only accuracy but also calibration.

approximately the same weight in the objective function. A common strategy is weight classes inversely proportional to their frequency (Wang et al., 2017). Recently, Cui et al. (2019) has proposed to reweight based upon the "effective" number of samples, which is defined per a mathematical formula. We here investigate if the calibration issues of an imbalanced dataset persist when using such mitigation strategies. Thus, we consider the standard cross-entropy (original), resampling inversely proportional to the frequency (sampling), reweighting inversely proportional to the frequency (weighted), the weighting scheme of Cui et al. (2019) (CB), and the focal loss of Lin et al. (2017) (focal). Additionally we consider label smoothing (Szegedy et al., 2016) (label smooth). We construct imbalanced datasets as in the previous section. Due to limited computational resources, we only consider the three smallest datasets for these experiments. Calibration errors are given in Table 1, and we give the largest and smallest calibration error among classes and the average ratio of the two. We see that issues of imbalanced calibration errors, while sometimes improving, still persist. Standard deviations are given in Table 2 in Appendix A.

**Label Quality.** When collecting labels, for example via crowdsourcing, a common issue is label quality (Patterson & Hays, 2012; Su et al., 2012; Callison-Burch & Dredze, 2010). For example, workers might have poor incentives to perform well or lack the necessary skills for quality labeling. To study the effects of potentially mislabeled data, we artificially inject symmetric noise into the training set. This is done by selecting a random subset of the training set corresponding to some fixed fraction, and then shuffling the labels of this set. This setup follows conventions in label noise literature (Patrini et al., 2017; Han et al., 2018). Given these noisy labels, we train the networks and evaluate them on the original test-set (which has no noise). We consider five levels of label noise in increments of $0.1$, starting at $0.0$. The resulting calibration errors for various noise levels are given in Figure 2, where we see that label noise increases the calibration error across all datasets. Additionally, we consider the effects of non-uniform noise, studied by e.g. Crammer et al. (2006). For class $i$, we linearly increase a noise level $p_i$ from $0.0$ to $0.5$. Classes are randomly ordered. For each image with original class $i$, with probability $p_i$ we assign it to a new random class. The reassignment probability to class $i$ is proportional to $p_i$. Results are given in Figure 3, where noisier classes suffer from worse calibration. This underscores how label quality control in e.g. crowdsourcing (Su et al., 2012) can be not only important for accuracy, but also for calibration of downstream models.

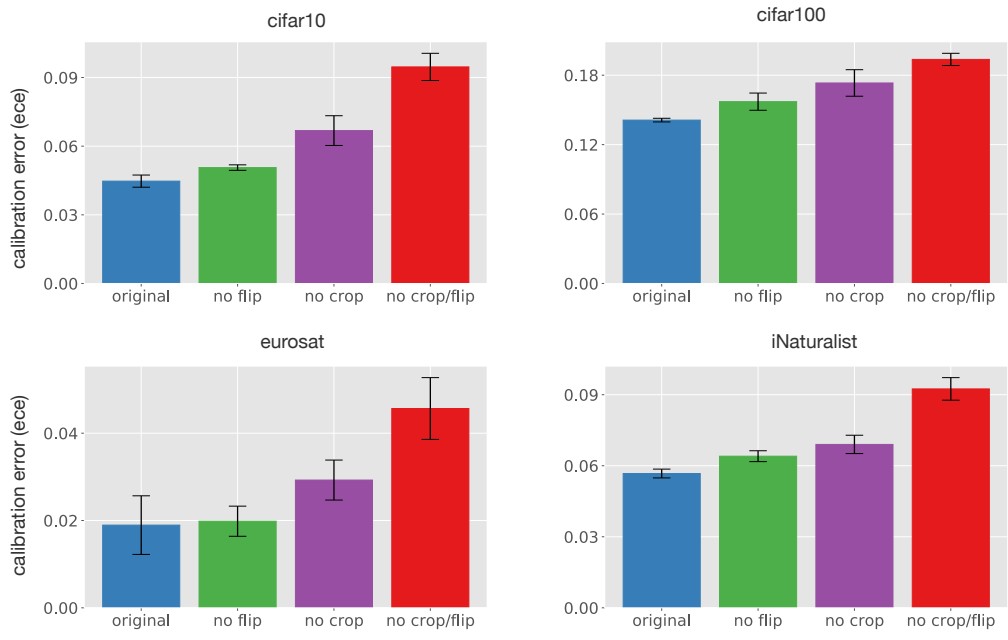

Figure 5: Calibration error under combinations of data augmentations. Following He et al. (2016), we consider randomized cropping and flipping. Removing these components, often used to artificially enlarge the dataset, increases the calibration error.

**Dataset size.** The perhaps most common concern for collecting data is the dataset size, and model accuracy typically grows with this size (Hestness et al., 2017). Crowdsourcing labels and bounding boxes for images is common practice, with many researchers investigating strategies to reduce needed queries (Su et al., 2012) In practice, dataset size can be limited by costs of labeling, but also by obtaining the actual data (Suram et al., 2017). Motivated by this, we study the effect of dataset size upon the calibration error. We simply subsample the training sets of the datasets uniformly at random and thereafter train on them, comparing different sizes of the resulting dataset. We consider subsampled sizes, measured in fractions of the original size, from 1.0 to 0.2 in increments of 0.2, and also consider 0.1. The test set is not subsampled. The results of these experiments are given in Figure 4. We see that smaller datasets have substantially larger calibration errors, demonstrating the dramatic effect that dataset size can have not only on accuracy, but also on calibration error.

**Augmentations.** Beyond actual dataset size, it is common to artificially increase the size of the dataset by augmenting it, e.g., randomly cropping the images or slightly shifting the color balance (Cubuk et al., 2018). We have seen that the size of the dataset influences calibration, and now consider the effect of increasing the effective dataset size via augmentations. We use both randomized cropping and horizontal flipping for our training and consider removing these components while keeping other training parameters fixed. The outcome of this experiment is shown in Figure 5, and we see that removing data augmentations significantly increases the calibration error. Viewing data augmentation as a strategy of extending the training set, we again see how smaller training sets increase the calibration error, just as in Figure 4.

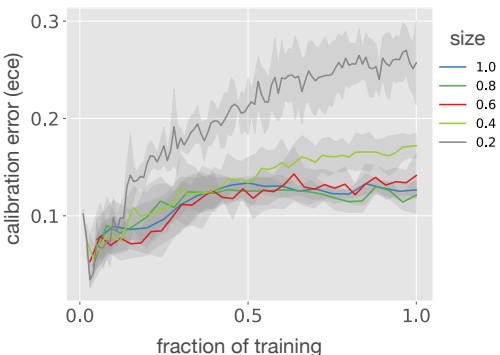

Figure 6: Calibration error of an NLP task during training for different dataset sizes. The dataset is subsampled, and we give the relative size.

**NLP.** While we primarily focus on computer vision, we here consider experiments in NLP. Language models often generate text via beam-search, and it has been observed that calibration is important for this process (Ott et al., 2018). Here we investigate the effect of dataset size on calibration in NLP. We considering translation of the IWSLT'14 German to English dataset (Cettolo et al., 2014) using transformer architectures (Vaswani et al., 2017) with code and default hyperparameters from the publicly available fairseq codebase (Ott et al., 2019). As before, we simply subsample the training set uniformly at random for variable sizes and train the transformer with all other training parameters fixed. Figure 6 shows how the mean calibration error, with standard deviations as error bars, varies during training. Again we see how the dataset size influences the calibration. It is natural to guess that there might be word-level calibration issues too, e.g., rare words might have worse calibration.

## 4  THEORETICAL MOTIVATION

Figure 4 and Figure 5 show that the size of the dataset affects calibration, with smaller datasets resulting in worse calibration. To explain this, let us consider the cross-entropy loss. We will let $l_{ij}$ denote the logit for image $i$ and class $j$. Furthermore, let $c_i$ be the index of the correct class for image $i$. The soft-max cross-entropy loss function is then defined as

$$\ell = \sum_i \ell_i = -\sum_i \log \frac{\exp(l_{ic_i})}{\sum_j \exp(l_{ij})} = -\sum_i \left( l_{ic_i} + \log\big(\sum_j \exp(l_{ij})\big) \right) \tag{2}$$

We note that this loss function decreases monotonically as the logit $l_{ic_i}$ increases. This implies that there is no global minimizer, but instead that if the other logits are fixed, we have $\ell_i \to 0$ as $l_{ic_i} \to \infty$. The logit tending to infinity implies that the *confidence* of the prediction tends to 1. The lack of minimizer for soft-max cross-entropy is in stark contrast with e.g. label smoothing, which penalizes large confidence, see Figure 7. Let us imagine that the network has infinite capacity. If we optimize it for a sufficient amount of time, we would expect $\ell$ to tend to zero, which implies that the logits tend to infinity. This corresponds to the confidence on the training set tending to $100\%$, which most likely implies overconfidence and *poor calibration*. We can formalize this observation, but we first need to state some assumptions.

**Assumption 1** *Let $\{x_i\} \in \mathbb{R}^{n \times d_x}$ be a dataset of $n$ datapoints with $d_x$ features each. Let $\{y_i\} \in \{0,1\}^{n \times c}$ be a one-hot label encoding that assigns each image one out of $c$ classes, where $c$ is a constant. We assume that all datapoints $\{x_i\}$ are distinct, i.e. $x_i \neq x_i, \forall i \neq j$.*

Under such assumptions, recent results in network expressivity say that one can essentially memorize a training set (Yun et al., 2019) if the width is at least of order $\mathcal{O}(\sqrt{n})$. In an idealized setting, where we optimize the function without computational considerations, such expressivity means that the loss function can be optimized towards its minimizer $0$. This means train set confidence growing to $100\%$, likely translating to poor calibration. We formalize this line of argument in Theorem 1.

**Theorem 1** *Let Assumption 1 hold. Let $f$ be a Relu networks with four or more layers and with width at least $\Omega(\sqrt{n})$ and parameters $w$. Let $\ell$ be equal to the loss function in eq. (2). Then (i) $\min_w \ell(f(w))$ no global minima; (ii) the confidence tends to 1 as $\ell \to 0$.*

The formal proof is given in Appendix B, we here provide some intuition. To prove $(i)$ and $(ii)$, it suffices that the network can fit the training set with $1.0$ accuracy. This is typically the condition in practice (Zhang et al., 2016), and whereas we consider an idealized argument without computation costs, the conclusions agree with our experimental results. For the sake of contradiction, let us assume that we are in a global minima with parameters $w$ and $100\%$ accuracy. Now consider $\ell_i$ when we scale the final layer by $(1+\alpha)$ for $\alpha > 0$. The network output is then $(1+\alpha)l_{ij}$, and $\ell_i$ is $-\log\big(\exp((1+\alpha)l_{ic_i})/\sum_j \exp((1+\alpha)l_{ij})\big) = \log(1 + \sum_{j \neq i} \exp\big((1+\alpha)(l_{ij} - l_{ic_i})\big)$. The fact that we have perfect train accuracy means that $(l_{ij} - l_{ic_i}) < 0 \ \forall j \neq c_i$. Thus, the loss must shrink, as $\sum_{j \neq i} \exp\big((1+\alpha)(l_{ij} - l_{ic_i})\big)$ decreases with $\alpha$ and as $\log$ is monotone. By contradiction, we are not in such a global minima. The results of Yun et al. (2019) say that we can always find weights which achieves perfect accuracy using $\mathcal{O}(\sqrt{n})$ parameters, and thus that there are no issues with fitting that dataset that prevents the loss from tending to 0. Thus, if the dataset is small compared to the number of parameters, we expect overconfidence and poor calibration. This conclusion agrees with observations of Guo et al. (2017), who show that depth and width increases miscalibration.

## 5 RELATED WORK

There is much recent work on how datasets influence the behavior of neural networks. Tsipras et al. (2020) shows how the process used to collect labels for imagenet can introduce bias into the resulting dataset. Miller et al. (2020) studies how the shift between different datasets can influence the performance of question and answering systems. Recht et al. (2019) construct new test sets for imagenet and cifar10, and observe differences in generalization compared to the original test sets. Imbalanced datasets is a common issue when applying machine learning in practice (Van Horn et al., 2018; Krishna et al., 2017; Thomee et al., 2016), and researcher often describe the "heavy-tail" of class labels (Cui et al., 2019). Traditional work on class imbalance includes Japkowicz & Stephen (2002) which investigates different sampling strategies, applicable to most machine learning models. For models of empirical risk minimization, one can instead reweight samples. A relatively recent reweighting scheme is proposed by Cui et al. (2019), where one uses the effective number of samples, which can be calculated from a simple formula.

For generating datasets, a common strategy is to employ crowdsourcing, where one lets ordinary people assign labels in a large-scale automated fashion, commonly via Amazon's Mechanical Turk system (Keith et al., 2017). Typical applications of crowdsourcing include analyzing images and providing bounding boxes (Patterson & Hays, 2012; Su et al., 2012), providing linguistic annotations for natural language (Callison-Burch & Dredze, 2010), or evaluating the relevance of search engines results (Alonso, 2013). Another application is machine learning debugging Ribeiro et al. (2016). The idea of eliciting and aggregating crowdsourced labels efficiently has inspired much algorithmic work (Khetan & Oh, 2016; Zhang et al., 2014). Common issues include finding tasks that result in high-quality labels, dealing with inconsistent labels (Karger et al., 2011; Zheng et al., 2017) and heterogenous workers (Ho et al., 2013).

Calibration in machine learning has been studied for a long time (Zadrozny & Elkan, 2001; Naeini et al., 2015) due to its practical implications. For neural networks, Caruana et al. (2015) demonstrated that shallow neural networks can yield well-calibrated predictions on classification tasks. In contrast, Guo et al. (2017) show how modern neural networks are ill-calibrated, with width and depth resulting in worse calibration scores, and investigate mitigation strategies. Neural network calibration has implications in NLP (Ott et al., 2018), fairness (Pleiss et al., 2017) and reinforcement learning (Kuleshov et al., 2018). For applications such as medicine (Miner et al., 2020), meteorology (Ren et al., 2015) and autonomous vehicles (Bojarski et al., 2016) it can be important for performance. Reliable uncertainty estimates also allow one to integrate DNNS with other probabilistic models, incorporating e.g. camera information (Kendall & Cipolla, 2015).

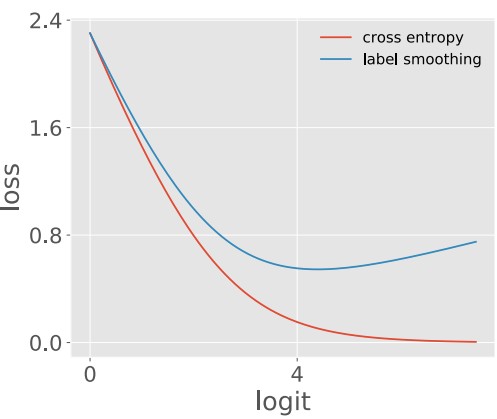

Figure 7: The softmax-cross entropy and label smoothing as a function of the logit of the correct class (other logits are zero). Cross-entropy decreases monotonically, resulting in large logits after optimization.

## 6 CONCLUSIONS

We have investigated the effects that datasets can have on network calibration. By generating label noise and class imbalance synthetically, we show how calibration error increases with label noise and few samples. We also study how calibration changes with dataset size. Our work points towards the importance of high-quality dataset curation for generating well-calibrated predictions, and highlight issues that are relevant in high-stakes applications such as autonomous vehicles and medical applications. These calibration issues can potentially be mitigated both at dataset curation time and training time; we defer such studies to future work. A practical takeaway from this work is that for sensitive applications, one should evaluate calibration when collecting datasets.

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

Table 2: Standard deviations corresponding to Table 1. We round to two digits.

| | cifar10 | | | cifar100 | | | eurosat | | |
|---|---|---|---|---|---|---|---|---|---|
| exp | min | max | ratio | min | max | ratio | min | max | ratio |
| original | 0.02 | 0.03 | 0.84 | 0.02 | 0.02 | 0.13 | 0.01 | 0.04 | 2.32 |
| sampling | 0.02 | 0.04 | 0.83 | 0.01 | 0.02 | 0.07 | 0.01 | 0.07 | 1.71 |
| weighted | 0.01 | 0.01 | 0.44 | 0.03 | 0.06 | 0.14 | 0.01 | 0.02 | 0.34 |
| label smooth | 0.01 | 0.04 | 1.06 | 0.0 | 0.01 | 0.03 | 0.0 | 0.05 | 0.48 |
| focal (Lin et al., 2017) | 0.02 | 0.03 | 0.99 | 0.03 | 0.01 | 0.13 | 0.01 | 0.05 | 1.3 |
| CB (Cui et al., 2019) | 0.01 | 0.04 | 0.61 | 0.01 | 0.04 | 0.15 | 0.01 | 0.02 | 0.32 |
| | cifar10 | | | cifar100 | | | eurosat | | |
| step | min | max | ratio | min | max | ratio | min | max | ratio |
| original | 0.01 | 0.02 | 4.37 | 0.02 | 0.01 | 0.56 | 0.01 | 0.07 | 5.53 |
| sampling | 0.01 | 0.03 | 2.82 | 0.02 | 0.01 | 0.38 | 0.0 | 0.05 | 5.7 |
| weighted | 0.01 | 0.05 | 1.18 | 0.01 | 0.04 | 0.39 | 0.03 | 0.03 | 0.98 |
| label smooth | 0.0 | 0.04 | 0.48 | 0.0 | 0.02 | 0.19 | 0.0 | 0.02 | 0.26 |
| focal (Lin et al., 2017) | 0.01 | 0.05 | 4.74 | 0.01 | 0.02 | 0.48 | 0.01 | 0.04 | 6.33 |
| CB (Cui et al., 2019) | 0.01 | 0.07 | 1.75 | 0.01 | 0.06 | 0.49 | 0.01 | 0.04 | 0.59 |

## A  APPENDIX

**Data Preprocessing.** For the cifar10 and cifar100 datasets, we use the original dataset size of 32-by-32 pixels. Cropping is performed by first padding with 4 pixels on each side, and thereafter performing a random 32-by-32 crop. For the eurosat dataset, the images are subsampled to 32-by-32 pixels. Since the eurosat dataset does not have a dedicated train/test split, we split it ourselves, using a fixed random tenth of the dataset for testing across all experiments. Random cropping is performed as for cifar. For the inaturalist dataset, the images are resized into 64-by-64 images, and we pad by 8 pixels on each side before extracting a random 64-by-64 crop.

Table 3: Hyper-parameters for computer vision.

| Parameter | Value |
|---|---|
| init. learning rate | 0.1 |
| learning rate decay per step | 0.1 |
| decay after | $\{50\%, 75\%\}$ |
| SGD momentum | 0.9 |
| batch size | 128 |
| horizontal flipping | True |
| cropping | True |
| weight decay | 0.0001 |
| loss | cross-entropy |

Table 4: Hyper-parameters used for Transformers (Vaswani et al., 2017). The architecture is "transformer-iwslt-de-en" of Fairseq (Ott et al., 2019).

| Parameter | Value |
|---|---|
| learning rate | 0.0005 |
| $\beta_1, \beta_2$ | 0.9, 0.98 |
| $\epsilon_{adam}$ | 0.00000001 |
| batch size | 32 |
| label smoothing | 0.1 |
| dropout probability | 0.3 |
| max-tokens | 4096 |

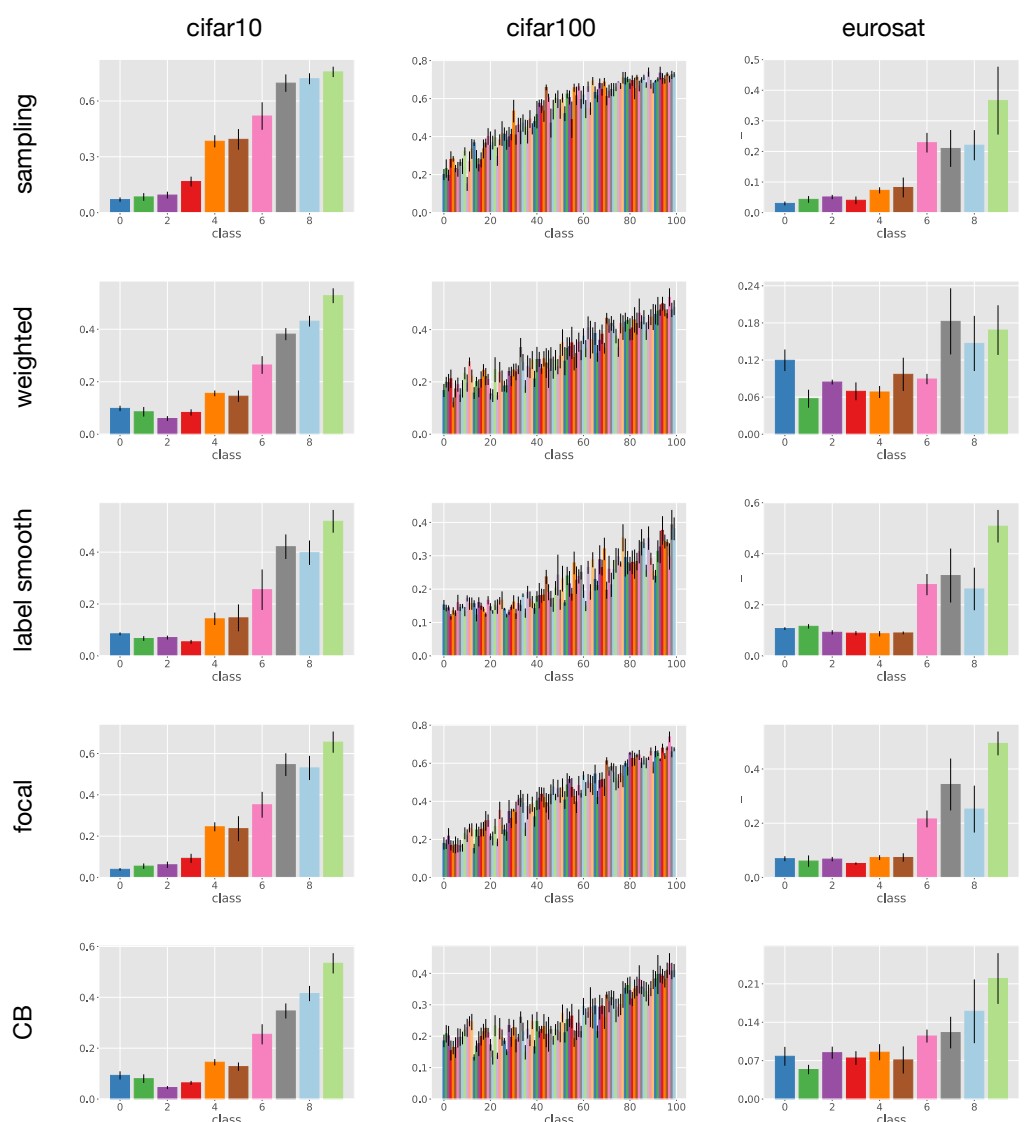

Figure 8: Classwise calibration error for all methods for exponential-inbalanced datasets. Class shuffling uses the same seed between methods.

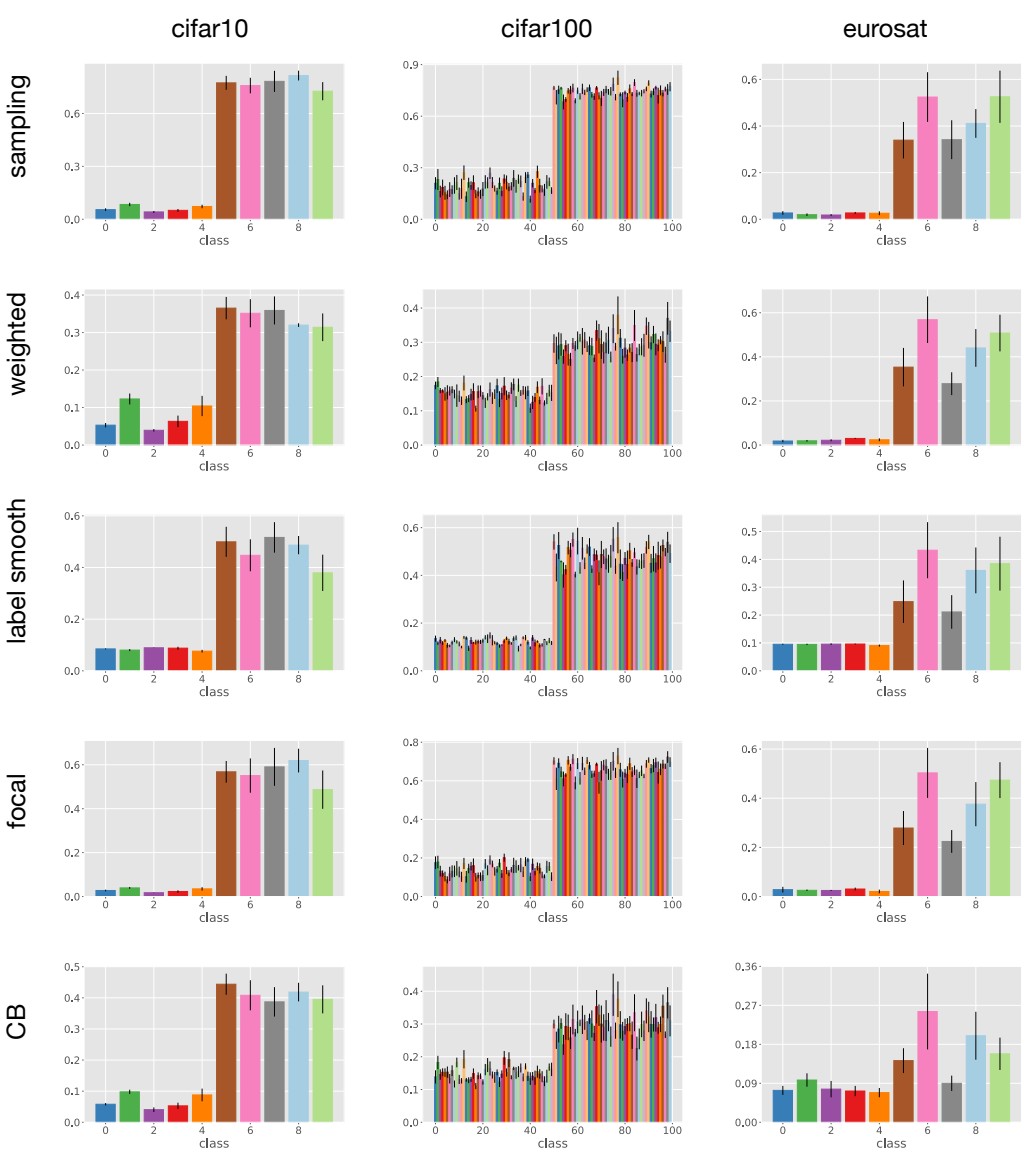

Figure 9: Classwise calibration error for all methods for step-imbalanced datasets. Class shuffling uses the same seed between methods.

# B    PROOF

The results of Yun et al. (2019) mean that under Assumptions 1, there exists a four-layer Relu network with $\Omega(\sqrt{n})$ parameters that can memorize the dataset, perfectly matching the one-hot encoding of the labels. For details, we refer to Yun et al. (2019).

**Proof Sketch for Theorem 1.** We start with part **(i)**. Assume for contradiction that we are in a global minima with loss $\ell_0$ and perfect accuracy. Now, consider $\ell_i$ when we scale the final layer by $(1+\alpha)$ for $\alpha > 0$. The network output is then $(1+\alpha)l_{kj}$, and $\ell_i$ is $-\log\left(\exp((1+\alpha)l_{ic_i})/\sum_j \exp((1+\alpha)l_{ij})\right) = \log(1+\sum_{j\neq i}\exp\left((1+\alpha)(l_{ij}-l_{ic_i})\right)$. The fact that we have perfect train accuracy means that $(l_{ij}-l_{ic_i}) < 0 \,\forall j \neq c_i$. Thus the loss must shrink, as $\sum_{j\neq i}\exp\left((1+\alpha)(l_{ij}-l_{ic_i})\right)$ decreases with $\alpha$ and as log is monotone. This contradicts us being in such a global minima. If we do not have perfect accuracy, there must be some example $k$ such that $\ell_k$ must be larger than $\log 2$ (since there is another class with larger logit). Results of Yun et al. (2019) says that we can find network with perfect accuracy and thereafter scale the weight to less than $\frac{\log 2}{n}$. **(ii)**. The confidence for a single example is $\frac{\exp(l_{ic_i})}{\sum_j \exp(l_j)} = \exp(-\ell_i)$. We have noted how $\ell \to 0$, since $\ell = \sum_i \ell_i$ we have $\ell_i \to 0$. This implies that $\exp(-\ell_i) \to 1$ which means that the confidence approaches 1. ∎

