# OpenReview forum: "Dataset Curation Beyond Accuracy"
_ICLR.cc/2021/Conference — Reject_

### Official Review · AnonReviewer3 · 2020-10-26
**This paper studied how dataset properties, included label quality and data size, affects calibration. The author  analyzed the affection of different dataset properties by testing on varying computer vision datasets  qualitatively. The experimental results show that poor calibration error accompanies with large noisy label rate, large imbalance ratio and small datasets. The author also provided the theoretical proof of why small data size results in high calibration error.**

**Rating:** 4
**Confidence:** 4

**Review:**

	This paper discussed how data properties (e.g., label noise, label imbalance, data size) affects calibration error. The author designed experiments on varying computer vision datasets (i.e., cifar10, cifar100, eurosat and iNaturalist) qualitatively: 1) calibration error for various individual classes under class-imbalance situation; 2) calibration error for different scale of label noise; 3) calibration error under non-uniform noise; 4) calibration error under various scale of dataset size; 5) Calibration error under different combinations of data augmentations. The experimental results show that poor calibration performance accompanies with large noisy label rate, large imbalance ratio and small dataset size. For the reason of small dataset size causing poor calibration error, this paper provided the theoretical proof.

	Advantages:
		○ The idea of considering a softmax-cross entropy logit loss to help explain how data size affect the calibration error is interesting.

	Major concerns:
		○ Organization should be improved. In particular, the factors that affect the calibration error should be listed and well described in a separate section (e.g., Intro -> Background -> (affected data properties) -> experiments), and the theoretical motivation could also be integrated in such section rather than put it after experiments.
		○ The novelty and practicability of this paper is limited, since this paper only tells people that low label quality and small data size would arise calibration error, the paper analyzed the factors qualitatively but not quantitatively. In the future research, the researcher still hard to justify how much calibration error the current dataset whould bring or can't tell whether the current the current classifier whould be robust enough to defense the calibration bring by the current set. An example is:
		[1] "Robustness of classifiers: from adversarial to random noise." Fawz et al. NIPS2016. This paper analyzed the robustness of classifiers quantitatively with considering adversarial and random noise.
	Minor comments:
		○ Table 1, "exp-inbalance" -> "exp-imbalance"
		○ Should the captions of Figure 2 and Figure 3 be changed?
		○ Assumption 1, "x_i != x_i" -> "x_i != x_j"
		○ Equation 2, "-sum_i(A+B)" -> "sum_i(A-B)".
		○ There are many typos in this paper, should go over the paper again and correct these small mistakes.

---

### Official Review · AnonReviewer4 · 2020-10-28
**not innovative, yet important simulations to help develop more robust experiments**

**Rating:** 6
**Confidence:** 3

**Review:**

In this work, authors demonstrate that dataset properties can significantly affect calibration and suggest that calibration should be measured during dataset curation. In the field of applied AI to real-life problem, we face all the time decision-makings on what is the most effective strategy in the pipeline (eg. sampling, noise, labeling) and this paper present some evidence for those decisions.


This type of work is important to systematically highlight areas or processes to follow in model development.
The study is not very novel, but important. Since the conclusions are very important and have key implications, I would suggest to apply this to more datasets, and also some of the existing synthetic datasets. Personally, I would like to see if these observations remain solid with more datasets and more variation of datasets.

I did not found any inconsistencies.

---

### Official Review · AnonReviewer2 · 2020-10-28
**Interesting work but can be extended**

**Rating:** 4
**Confidence:** 3

**Review:**

This work is an empirical survey of the calibration problem with convnets. The authors use several existing benchmark datasets and create synthetic class-imbalance for datasets that are initially balanced. They then extend the well-known results on higher prediction error of minority class, to its calibration error. The work investigates several existing methods that alleviate prediction error in imbalanced datasets and examine their effect on calibration error. At last, the effect of dataset size and data augmentation on calibration error is reported. Later on, the effect of random label noise is also examined. The observations, although not surprising, have not been reported before
The work is interesting, the writing is clear, and the experiments are comprehensive. Although the observations are very informative, the overall contribution of the paper is not sufficient for the ICLR venue. The work is mostly focused on reporting an existing issue with no major theoretical analysis of the problem and guidelines for alleviating the mentioned problems. The paper is in an interesting direction but needs to become more mature.

Questions and suggestoins:
1- The label noise experiments are interesting. In reality, label noise is rarely random and is structured. It would be more helpful if the authors could extend the experiment to incorporate such scenarios.
2- There seems to be an interesting difference among various reweighting methods in Table 1. It would be interesting if authors compared their calibration error performance to their prediction error performance to find out if there is a trade-off or the two phenomena are in the same direction.
3- In a lot of experiments, for instance the dataset size, it's expected to have higher calibration error for smaller data. It would be more informative if the general trend of calibration error is compared with the trend in prediction error side by side.

---

### Official Review · AnonReviewer1 · 2020-10-29
**If we were reporting accuracy in the paper experiments, how different would the conclusions be?**

**Rating:** 4
**Confidence:** 4

**Review:**

The paper is an empirical study looking at how different dataset properties affect model calibration in the context of vision tasks. All experiments use a specific well-known vision model (ResNet 50).

In particular, the dataset properties that are investigated are:

- Balanced/Unbalanced classes.
- Label quality.
- Dataset size.
- Augmentations.
- NLP.

I briefly present the main conclusions below.

- Balance in classes. Often times some classes have way more datapoints than others. The authors look at four datasets (Cifar 10, Cifar 100, Eurosat, iNaturalist). The last one's classes are unbalanced, whereas the first three require some sampling method to (artificially) make them unbalanced (note in this case by design there is no relationship between balance/unbalance and the class properties). Figure 1 shows the results. For Cifar and Eurosat those classes with more examples are better calibrated. The trend is somewhat similar for iNaturalist.

Then, the authors present a number of approaches people have tried in the past to mitigate the consequences of unbalance in data. They repeat the previous experiment (on Cifar 10, Cifar 100, Eurosat) but, this time, using each of those methods while training the model. Table 1 shows the results. The ratio column offers very mixed results depending on the dataset and method. The authors conclude that overall the imbalance in calibration persists in most cases.

Q. How do these results compare to accuracy? One would also expect to do better on classes with more data.

- Label quality. The authors tackle the question of how label noise affects calibration. In order to do that, they artificially inject noise to the "true" labels with increasing probability. Figure 2 summarizes the calibration error for a number of datasets and noise level. The pattern is clear: the more noise, the worse the calibration. Importantly, the calibration is measured on a test set that is not perturbed with random noise. Accordingly, results were to be expected: there's a mismatch between training and test distributions, and the further apart they are, the less "meaningful" predicted probabilities one should expect. Again, it would be informative to see how the *accuracy* of the model also degrades under this circumstances. Similarly, Figure 3 shows the effect of non-uniform noise across classes. Those classes "attacked" with more noise are worse calibrated.

- Dataset size. Another important practical aspect to study is dataset size. The authors subsample uniformly at random a fraction of the data points, and measure ECE. Figure 4 shows how models trained on more data are better calibrated. Again, the accuracy of the model should also be shown for context.

- Augmentations. It is common to use data augmentation to train better models; augmentations make the effective datasize larger. Figure 5 shows how removing augmentation axes leads to worse calibration. The same probably applies to accuracy (that's the reason why people use this!). This result is probably intimately related to the previous point (dataset size).

- NLP. The conclusions regarding dataset size also hold with a Transformer on an NLP dataset.

Finally, Section 4 provides some theoretical explanation. We can summarize this as: the cross-entropy loss wants to have more and more confidence / probability on the right class for a given example, and when the data is small and the model powerful enough, we can basically memorize it to make cross-entropy happy. This, however, leads to overconfidence and poor calibration.

On one hand, it's recently becoming clear that ECE is not a very robust estimator. Depending on design choices (as number of bins, argmax vs all, adaptive versus fixed bins, etc.) the ranking among models and conclusions can change wildly [1]. On the other, this study fixed a specific model, so one could say that the conclusions are "shown" for the (dataset, model) pairs. Still, I believe the conclusions are true in a more general setting, though, and the model is fairly reasonable. However, while the paper is titled "Dataset Curation Beyond Accuracy", I do not see how the outcome and conclusions of all these experiments would be different if we were looking at accuracy rather than calibration. The authors should measure, include, and address this, and try to disentangle both aspects, or argue for any correlation / causation relationship among them.

[1] - Measuring Calibration in Deep Learning - https://arxiv.org/abs/1904.01685

---

### Author Response · Authors · 2020-11-23
**Rebuttal**


We thank the reviewers for their thoughtful comments and ideas to improve the paper. We address major questions from reviewers (R1, R2, R3, R4) below.



# R1.

----------------

How do these experiments relate to accuracy?

- There is no simple relationship between calibration and accuracy. It has previously been shown that models with more parameters have worse calibration despite being more accurate. We show how e.g. smaller datasets result in worse calibration in addition to worse accuracy. Generally, noisy classes have worse accuracy and smaller datasets result in worse accuracy. There is plenty of work on what factors influence accuracy, and we have specifically focused on calibration to complement previous studies and highlight another way in which datasets can influence training.




# R2.
----------------

It would be helpful to consider non-synthetic noise

- This is a great point, we’d love to do that but in order to measure the noise, it’s preferable to have “correct” ground truth labels to be able to assess the noise rate. We’re happy for any pointers towards specific datasets that could be used for this.




# R4.
----------------

The organization can be improved by putting experiments last

- This is a good point, the suggested layout of stating the main observations and the theoretical motivation first, and thereafter going through the experiments would likely improve the clarity of the paper. Thanks!


The  practicability is limited

- We believe that the first step towards addressing an issue is being aware of its existence. To the best of our knowledge, the role that datasets can play in calibration has not been observed before, thus making the community aware of this issue is a first step towards developing methods to address it. Furthermore, the results of this paper might inform practitioners on the importance of dataset curation even when target accuracy is met.

---

### Decision · Program_Chairs · 2021-01-07
**Final Decision**

**Decision:**

Reject

**Comment:**

The authors empirically analyse the properties of datasets which lead to poor calibration. In particular, they show that high class imbalance, high degree of label noise, and small dataset size are all likely to lead to poor overall calibration or poor per-class calibration. While there are some interesting insights in this work, the reviewers argued that the contribution is not substantial enough for ICLR. To improve the manuscript the authors should consider accuracy and calibration jointly and extend the results pertaining to label noise which were appreciated by the reviewers. For the former, the same conclusions hold for accuracy, instead of calibration, which raises the question of their relationship -- is there a tradeoff? For the latter, the reviewers pointed to a concrete extension with structured label noise. Finally, the theoretical analysis is a step in the right direction, but the assumption on the width of the network required to fit the training set is too restrictive in practice. Therefore, I will recommend rejection.